# A Two-Stage Optimal Scheduling Model of Microgrid Based on Chance-Constrained Programming in Spot Markets

**Jiayu Li [1], Caixia Tan [1], Zhongrui Ren [2], Jiacheng Yang [1], Xue Yu [1] and Zhongfu Tan [1,3,\*]**

[1] School of Economics and Management, North China Electric Power University, Beijing 102206, China; lijiayu1965@126.com (J.L.); cx_sp@ncepu.edu.cn (C.T.); yjcc1227@163.com (J.Y.); yx_2018n@163.com (X.Y.)

[2] Power Generation Department, Fujian Yongfu Power Engineering Co., Ltd., Fuzhou 350108, China; renzhongrui@fjyongfu.com

[3] School of Economics and Management, Yan'an University, Yan'an 716000, China

\* Correspondence: tanzhongfu@sina.com; Tel.: +86-010-6177-3118

**Abstract:** Aimed at the coordination control problem of each unit caused by microgrid participation in the spot market and considering the randomness of wind and solar output and the uncertainty of spot market prices, a day-ahead real-time two-stage optimal scheduling model for microgrid was established by using the chance-constrained programming theory. On this basis, an improved particle swarm optimization (PSO) algorithm based on stochastic simulation technology was used to solve the problem and the effect of demand side management and confidence level on scheduling results is discussed. The example results verified the correctness and effectiveness of the proposed model, which can provide a theoretical basis in terms of reasonably coordinating the output of each unit in the microgrid in the spot market.

**Keywords:** microgrid; spot markets; chance-constrained programming; demand side management

---

## 1. Introduction

Compared with the medium-term and long-term market, the spot market of electric power mainly includes electricity commodity trading with finer time granularity within a few days before and during the day [1], which can form a time-of-use electricity price reflecting the real production cost of commodities and represent the actual situation of power supply and demand. Thus, it can improve the timeliness of power resource allocation and play a vital role in the power market system [2,3].

With the issuance of the Notice on Carrying out the Pilot Work of Power Spot Market Construction by the National Development and Reform Commission (NDRC), the construction of China's power spot market has officially started. As of June 2019, the first batch of eight power spot market pilot projects have all started their simulation test run, which indicates that the construction of a power spot market with Chinese characteristics has achieved positive results. Under the background of accelerating the construction of a power spot market, research on how to guide the reasonable decision-making of transaction participants can be conducive to not only achieving the maximum economic benefits of each subject, but also the entire market's efficient allocation of resources, which can ensure the orderly economic operation of the spot market and has important theoretical significance and practical value [4–6].

The spot market transaction mainly includes two kinds of participants: the power generation subject and power consumption subject. As the traditional thermal power units are more involved in the peak load regulation auxiliary service market [7], the research on spot market decision-making for power generation focuses on renewable energy producers. However, renewable energy generation

has the characteristics of being intermittent and uncertain, which leads to the failure of renewable energy generation companies to formulate transaction declaration strategies by the declaration model of conventional power generation enterprises. In view of the above problems, the literature [8] assumes that wind power generation enterprises are the price receivers in the spot market, and fully considering the uncertainty of the wind turbine (WT) output and clearing price, the optimal output model of wind power plants by using stochastic simulation technology can be constructed. In [9], based on the study in [8], a penalty coefficient was introduced to punish the deviation between the real-time market output and daily market declared volume. Taking the maximum total profit as the objective function, a stochastic optimization model of wind power business in the two-stage market was constructed. This model fully considers multiple market stages and uncertain factors, further optimizing the declaration strategy of wind power suppliers.

In addition, with the progress of energy storage technology, a variety of energy storage system (ESS) has been continuously studied to reduce the renewable energy output's uncertainty. In [10,11], by virtue of the ESS's flexible charging and discharging function, the coupling between the ESS and the wind farm was realized. By establishing the linear affine function between the electricity market price, the wind power predicted output deviation, and the power of the ESS charge and discharge, the stochastic optimization declaration strategy of WT was proposed. In [12], considering both the randomness of the wind power output and the uncertainty of the spot price, a hybrid power plant's two-level stochastic optimization model consisting of wind power suppliers and electric vehicles was proposed, which could effectively reduce the bidding risk and increase the profit of the hybrid power plant.

In research on the spot market decision-making of consumers, the main goal is to minimize the purchase cost of the electricity sellers or large users. In [13], a short-term planning model was proposed to predict the load curve under the real-time electricity price, which fully considers the fluctuation of the user load and market electricity price. A day-ahead decision-making model of the electricity sellers based on stochastic optimization was established. In the study by [14], based on [13], the demand response of sensitive users was introduced, considering the uncertainty of the spot price and the consumer behavior of users. The day-ahead market quotation strategy model of selling by the electricity supplier was constructed. In [15,16], based on robust optimization and stochastic mixed integer programming, the optimal power purchase decision-making and quotation decision-making models for the day-ahead market of selling by the electricity supplier were proposed.

The above research on the optimization decision of the spot market only involves a single subject and fails to form the link between the generation side and the power consumption side. As the link between the power load and distributed energy in a microgrid integrates multiple system resources, it can simultaneously carry out energy supply and demand response, realizing the efficient allocation of power resources, so it has unique advantages in the process of participating in the spot market. In [17], aiming at maximizing the real-time market profit of the microgrid, each unit of the microgrid's bidding strategies were studied under two bidding mechanisms. However, the demand response of the microgrid load was not considered and the users' potential was not fully explored. In [18], the price response model of a sensitive load was introduced. Aimed at minimizing the expected net cost, a hybrid stochastic robust optimization model was proposed. However, the penalty cost caused by an unbalanced power in the real-time market of microgrid was ignored, and research on the two-stage market transaction of day-ahead and real-time is lacking. To sum up, there have been a few studies on microgrids participating in spot market transactions, and there are still some deficiencies.

Compared with the existing research, the innovation of this paper lies in the following points. First, in the microgrid spot market decision-making issues, it fully considers the fluctuation of the stochastic micro-source output and spot market electricity price and discusses the economics and reliability of simultaneous optimization on both sides of supply and demand. Second, it employs stochastic simulation technology to transform the optimal scheduling problem into a problem with maximum system revenue under a certain confidence level and also uses the particle swarm optimization (PSO)

algorithm to solve it effectively. Third, it optimizes the microgrid scheduling by changing the objective function and constraining the confidence level to meet the decision-makers of different risk preferences.

　　The rest of this manuscript is organized as follows. In Section 2, considering the uncertain output of the WT and photovoltaic (PV), a microgrid energy management system integrating wind–PV–load-storage was established, which is the technical basis for building a two-stage scheduling model of a microgrid. Section 3 discusses the process of the microgrid participating in the spot market, according to the operation rules of China's power spot market and microgrid construction methods. On this basis, a microgrid two-stage optimal scheduling model based on the stochastic chance constraint was constructed, which takes the predicted power of renewable energy generation and the predicted price of the spot market as random variables. In Section 4, the stochastic simulation technology is used to deal with the random variables in the model, and then a PSO algorithm suitable for solving this model is proposed. Section 5 introduces the simulation results, verifies the validity of the model, and discusses the effect of demand side management and confidence level on scheduling results. Section 6 highlights the main conclusions of this paper.

## 2. Microgrid System Model Considering Randomness of Power Supply Output

　　A microgrid is a small-scale power generation and distribution system including distributed power supply, user load, ESS, and control system. Each unit is connected to each other by power electronic devices and is connected to the public power grid through static switches, which can realize a flexible conversion between the grid connection and grid disconnection. The structure of the microgrid system is shown in Figure 1.

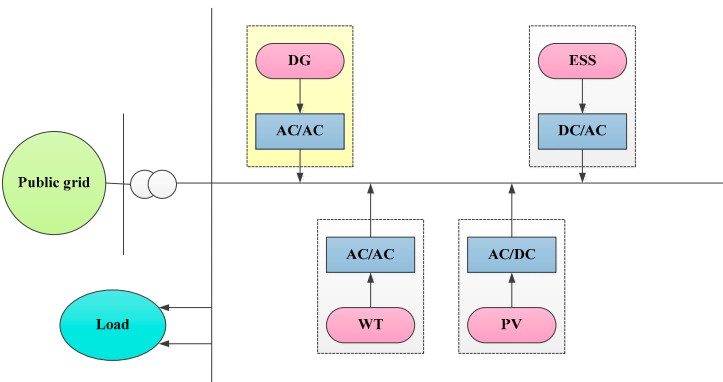

**Figure 1.** The structure of the microgrid system.

　　As shown in Figure 1, random micro-sources with uncertain output such as WT and photo voltaic are given priority to generate power to provide the electrical energy required for internal loads. ESS improves the power supply rejection ratio of random micro-sources through low storage and high-discharge, and controllable micro-sources such as diesel generator (DG) provide backup services when random micro-sources and the ESS cannot meet the load demand.

*2.1. Stochastic Power Supply Output Model*

2.1.1. Stochastic Output Model of Wind Turbine

　　Due to the influence of wind energy quality in different periods, the output power of WT often tends to be intermittent. Through statistical research on a large number of measured data, the two-parameter Weibull distribution is widely used to fit the distribution of the actual wind speed [19], and the probability density function can be expressed as:

$$f_{WT}(v) = \frac{k}{c}\left(\frac{v}{c}\right)^{k-1} e^{-\left(\frac{v}{c}\right)^k} \tag{1}$$

where $v$ is the wind speed; $k$ is a shape parameter, and its value fluctuates in the range of 1.8–2.8 according to the specific situation of wind energy resources; and $c$ is a scale parameter reflecting the average wind speed during the period.

The relationship between the output power of the WT and the actual wind speed $v$ is as follows:

$$P_{WT} = \begin{cases} 0 & v < v_{ci}, v \geq v_{co} \\ \frac{v^3 - v_{ci}^3}{v_r^3 - v_{ci}^3} p_r & v_{ci} \leq v < v_r \\ p_r & v_r \leq v < v_{co} \end{cases} \tag{2}$$

where $P_{WT}$ is the output power of the WT; $v_{ci}$ and $v_{co}$ represent the cut-in wind speed and the cut-out wind speed, respectively; $p_r$ is the rated power of the WT; and $v_r$ is the rated wind speed.

### 2.1.2. Stochastic Output Model of Photovoltaic

Like wind energy resources, the light and temperature received by PVs will constantly change due to the characteristics of solar radiation distribution, which results in the great uncertainty regarding its output. Related research shows that the solar illumination intensity approximately obeys a Beta distribution in a certain period of time [20]. Its probability density function can be expressed as:

$$f_{PV}(l) = \frac{\Gamma(a_l + b_l)}{\Gamma(a_l)\Gamma(b_l)} \left(\frac{l}{l_{max}}\right)^{a_l - 1} \left(1 - \frac{l}{l_{max}}\right)^{b_l - 1} \tag{3}$$

$$\Gamma(x) = \int_0^\infty e^{-t} t^{z-1} dt \quad z > 0 \tag{4}$$

where $l$ is the solar irradiance; $a_l$ and $b_l$ are the shape parameters; and $l_{max}$ is the maximum irradiance.

The relationship between the output power of the PV unit and the solar irradiance can be expressed as:

$$P_{PV} = l S_{PV} \eta \tag{5}$$

where $P_{PV}$ is the output power of PV units; $S_{PV}$ is the total area of PV units; and $\eta$ is the photoelectric conversion efficiency of PV units.

### 2.2. Controllable Power Supply Output Model

Distributed power generation with a controllable output is generally composed of small power generation equipment such as DG and a gas engine generator, and has the advantages of a fast start–stop speed (only 15 s) and stability. It often provides backup assistance for microgrid systems when the new energy unit and the ESS cannot meet the load demand. Therefore, its power generation is jointly determined by the user power load, the output of the new energy unit, and the ESS discharge power. By using linear function to simplify the operation cost of the controllable micro-sources and ignoring its climb and start–stop constraints [21], the following cost functions and constraints can be established:

$$C_c = \sum_{\tau=1}^{\omega} \sum_{n=1}^{N} \sum_{s=1}^{S} C_{c,n,s}^A Q_{c,n,s,\tau} \tag{6}$$

$$0 \leq Q_{c,n,s,\tau} \leq Q_{c,n,s}^R - Q_{c,n,s-1}^R \tag{7}$$

$$0 \leq Q_{c,n,1,\tau} \leq Q_{c,n,1}^R \tag{8}$$

where $\omega$ is the number of all-day spot market hours; $S$ is the total number of segments of piecewise linear operation cost; $C_c$ is the operating cost of the controllable unit; $N$ is the total number of controllable units; $C_{c,n,s}^A$ is the average power generation cost of unit $n$ at the segmentation of $s$; $Q_{c,n,s,\tau}$ is the power

generation capacity of unit $n$ at the segmentation of $s$ in the time period of $\tau$; and $Q^R_{c,n,s}$ is the rated power generation capacity of unit $n$ at the segmentation of $s$.

*2.3. Energy Storage System Model*

ESS can realize the energy transfer of a microgrid system through ESS, and release and optimize the microgrid power output curve to adapt to the price change in the spot market and ensure the economic operation of the microgrid. Under the current energy storage technology conditions, the microgrid ESS still mainly uses battery energy storage. The state of charge can be expressed as:

$$\begin{cases} SOC_\tau = SOC_{\tau-1} - \dfrac{Q_{b,\tau}}{\eta_{out} \times W_b} & Q_{b,\tau} \geq 0 \\ SOC_\tau = SOC_{\tau-1} - \dfrac{Q_{b,\tau} \times \eta_{in}}{W_b} & Q_{b,\tau} < 0 \end{cases} \tag{9}$$

where $SOC_\tau$ is the state of charge of battery energy storage in the time period of $\tau$; $Q_{b,\tau}$ is the charge and discharge amount of the ESS in the time period of $\tau$ ($Q_{b,\tau}$ is positive for battery discharge and negative for battery charge); $\eta_{in}$ and $\eta_{out}$ are the charge and discharge efficiencies respectively; and $W_b$ is the battery energy storage capacity. The charging and discharging constraints and power constraints of ESS [22] can be expressed as:

$$\begin{cases} SOC_{\min} \leq SOC_\tau \leq SOC_{\max} \\ -0.25W_b \leq Q_{b,\tau} \leq 0.4W_b \end{cases} \tag{10}$$

where $SOC_{\min}$ is the lower limit of the state of charge, preferably 0.2; and $SOC_{\max}$ is the upper limit of the state of charge, preferably 0.9.

The operating cost of the ESS can be expressed as follows:

$$C_b = \sum_{\tau=1}^{\varpi} \mu_b |Q_{b,\tau}| \tag{11}$$

where $C_b$ is the depreciation loss cost generated by battery energy storage's charging and discharging and $\mu_b$ is the battery loss coefficient.

*2.4. User Demand Response Model*

In order to avoid the uncertain risks brought by the spot market and realize the optimal allocation of power resources, a microgrid can use demand side management to encourage electricity users to change their electricity consumption behaviors in order to optimize the load curve. Among them, the TOU (time-of-use) price is one of the most effective ways to guide users to participate in load regulation. According to micro-economic principles, the relationship between electricity consumption and electricity price can be described by the elasticity of power price [23]. The transfer of electricity consumption by users due to participation in the TOU demand response can be expressed as:

$$Q'_\tau = Q^0_\tau \times \left\{ 1 + e_{\tau\tau} \times \frac{\left[P'_\tau - P^0_\tau\right]}{P^0_\tau} + \sum_{\substack{\varsigma = 1 \\ \varsigma \neq \tau}}^{24} e_{\varsigma\tau} \times \frac{\left[P_\varsigma' - P^0_\varsigma\right]}{P^0_\varsigma} \right\} \tag{12}$$

$$e_{\varsigma\tau} = \frac{\Delta Q_\varsigma / Q^0_\tau}{\Delta P_\tau / P^0_\tau} \tag{13}$$

where $Q^0_\tau$, $Q'_\tau$, $P^0_\tau$, and $P'_\tau$ are respectively the user's electricity consumption and electricity price in the time period of $\tau$ before and after the demand response; $P^0_\varsigma$ and $P_\varsigma'$ are the user's electricity price

at the hour of $\varsigma$ before and after the demand response, respectively; $e_{\tau\tau}$ and $e_{\varsigma\tau}$ are the self-elasticity and the cross-elasticity, respectively; and $\Delta Q_\tau$ and $\Delta P_\tau$ are the change of electricity consumption and electricity price in the time period of $\tau$, respectively.

The cost of implementing demand side management for a microgrid can be expressed as:

$$C_d = \sum_{\tau=1}^{\bar{\omega}} P_\tau^0 Q_\tau^0 - P'_\tau Q'_\tau \tag{14}$$

## 3. A Two-Stage Scheduling Model of a Microgrid Based on Chance-Constrained Programming in Spot Markets

### 3.1. The Mechanism of a Microgrid's Participation in Spot Market Trading

Based on the research on the operation scheme and the simulated trial operation of power spot markets in Guangdong, Zhejiang, and the Western Economic Zone of Inner Mongolia, China's power spot market operation pilot generally adopts the way of "day-ahead + real-time" for settlement. The day-ahead market starts on the day before the actual delivery of electricity, and power generation enterprises and power users sequentially declare the electricity price curve and the electricity demand curve. The electric power control center shall conduct Market clearing and settle according to the clearing results. The real-time market is the balance of forecasting errors in the day-ahead market, which is carried out before the actual delivery of electricity. The declaration of the market participants is similar to that of the day-ahead market. Finally, the power control center carries out real-time market clearing and settles the deviation power according to the real-time clearing price.

According to the Proposed Measures for Grid-connected Microgrid Construction issued by the State Energy Administration of China, a microgrid can be used as a power selling company with the distribution grid management right to participate in the spot market transactions of electric power. Therefore, the research object is the grid-connected microgrid that participates in the spot market transaction as the seller of electricity.

As the main seller of electricity, the microgrid participates in the spot market transaction including two settlement stages: day-ahead market and real-time market. The day-ahead market will be launched one day in advance, the microgrid will only declare the electricity, not price, and settle the electricity consumption curve according to the clearing result. Based on the actual operation results of the trading day, the real-time market will settle the deviation between the actual electricity consumption and the bid winning curve of the day-ahead market, according to the real-time price.

In addition, the State Energy Administration issued "the regulatory measures for the full purchase of renewable energy power by power grid companies", which requires power grid companies to purchase all the on-grid power of renewable energy power generation enterprises except for the market trading power, which means that the extra power of new energy used for spontaneous self-use can be sold to power grid companies.

Therefore, the energy interaction between the microgrid and power grid involved in spot market transactions mainly includes the daily market declared power, real-time market deviation power, and on-grid power. At the same time, the microgrid will generate operating costs and electricity sales revenue, which will affect the total revenue. The settlement process of the microgrid participating in spot market transactions is shown in Figure 2. It is necessary to fully consider the uncertainty of the spot market price and the randomness of the power supply output, and optimize and schedule various adjustable resources in the microgrid to maximize the benefits of participating in the spot market.

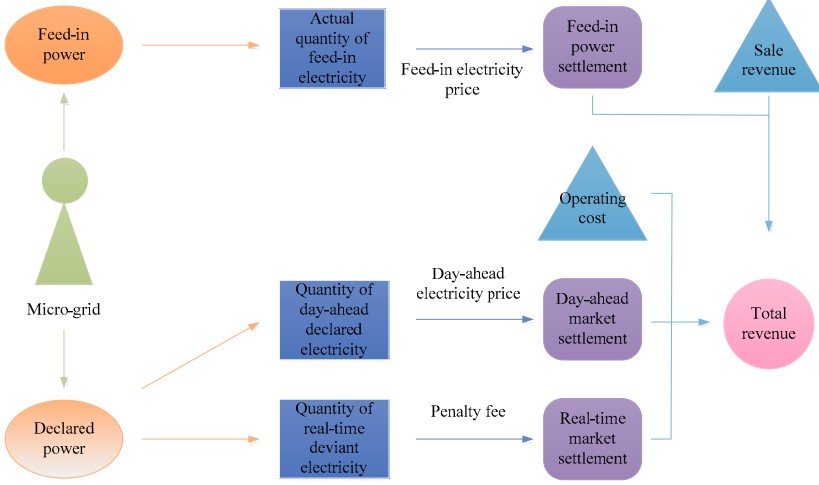

**Figure 2.** The flow chart of the microgrid participating in the spot market.

### 3.2. Microgrid Scheduling Model of Day-Ahead Market

#### 3.2.1. Market Clearing Price Forecast

Since the market clearing electricity price can be expressed in the form of an equal time series, the autoregressive AR (p) model is introduced to predict the spot market price. Based on the foreign electricity market transaction data and related scholars' research as well as the fact that the information criterion function can be used to select the order of the AR (p) model as 1, the clearing electricity price predicted by the day-ahead market can be expressed as:

$$P_{c,\tau}^{DA} = C(\tau) + \phi_1 P_{c,\tau}^{DA,D-1} + \varepsilon_\tau \tag{15}$$

where $P_{c,\tau}^{DA}$ and $P_{c,\tau}^{DA,D-1}$ are the market clearing electricity prices in the time period of $\tau$ in the day-ahead market in the current day and the previous day, respectively; $C(\tau)$ is the average value of the market clearing electricity price in the time period of $\tau$ in the day-ahead market; $\phi_1$ is the autoregressive coefficient; and $\varepsilon_\tau$ is the white noise sequence with a mean of 0 and a variance of $\sigma^2$.

#### 3.2.2. Microgrid Day-Ahead Scheduling Model Based on Chance-Constrained Programming

In order to participate in spot market transactions, microgrid needs to consider many uncertain factors to achieve economic scheduling. As an important branch of stochastic programming theory, chance constrained programming allows the optimal scheduling results not to meet the constraints to some extent, but requires the probability of the constraints to be established not less than a certain level of confidence. Thus, we can deal with the random variables in microgrid scheduling by softening the constraints, which can be expressed as follows:

$$\min \overline{f} s.t. \begin{cases} C_r\{f(x,\varepsilon) \leq \overline{f}\} \geq \alpha \\ C_r\{g_m(x,\varepsilon) \leq 0\} \geq \beta & m = 1,2,\cdots,p \\ g_n(x,\varepsilon) \leq 0 \end{cases} \tag{16}$$

where $x$ and $\varepsilon$ are the decision variables and the random variables, respectively; $f(x,\varepsilon)$ is the objective function; $g_m(x,\varepsilon)$ is the random chance constraint; $g_n(x,\varepsilon) \leq 0$ is the conventional constraint; $\alpha$ and $\beta$ are the confidence levels; and $\overline{f}$ is the expected value of the objective function, that is, the minimum value of the objective function when the probability is not lower than $\alpha$.

As for the microgrid, the day-ahead optimal scheduling embodies the maximum comprehensive expected income. In other words, the output of each system is optimized to form the optimal power demand curve for each period by accurately forecasting the load and the day-ahead clearing price in

order to achieve the maximum profit in the day-ahead market competition. Therefore, the following chance-constrained programming model can be established. The superscript *DA* of all symbols in the first stage optimization model represents the day-ahead market stage and the superscript *RT* of all symbols in the second stage optimization model represents the real-time market stage.

 (1) The objective function

$$\begin{cases} \max \overline{f_{DA}} \\ C_r\{f_{DA} \geq \overline{f_{DA}}\} \geq \alpha \end{cases} \tag{17}$$

$$f_{DA} = R_s^{DA} - C_c^{DA} - C_b^{DA} - C_d^{DA} \tag{18}$$

$$R_s^{DA} = \sum_{\tau=1}^{\omega} \left( P_{s,\tau} Q_\tau^{DA\prime} - P_{c,\tau}^{DA} Q_{r,\tau}^{DA} + P_{on} Q_{on,\tau}^{DA} \right) \tag{19}$$

where $R_s^{DA}$ is the market clearing income of the microgrid day-ahead market; $P_{s,\tau} Q_\tau^{DA\prime}$ represents the revenue from the electricity sale of the microgrid in the $\tau$ period (the object of electricity sale is the users); $P_{c,\tau}^{DA} Q_{r,\tau}^{DA}$ represents the power purchase expenditure of the microgrid in the day-ahead market in the $\tau$ period; $P_{on} Q_{on,\tau}^{DA}$ represents the proceeds from the sale of the remaining electricity of the microgrid to the grid company in the $\tau$ period (the object of sale is the grid company); $P_{s,\tau}$ and $Q_\tau^{DA\prime}$ are the users' electricity price and the predicted response electricity in the time period of $\tau$, respectively; $P_{c,\tau}^{DA}$ and $Q_{r,\tau}^{DA}$ are the market clearing electricity price and of the day-ahead market and the microgrid day-ahead declared power in the time period of $\tau$, respectively ($Q_{r,\tau}^{DA}$ is the electricity purchase demand declared by the microgrid in the day-ahead market of the day before the trading day); and $P_{on}$ and $Q_{on,\tau}^{DA}$ are the microgrid on-grid electricity price and the predicted on-grid power in the time period of $\tau$, respectively.

 (2) Constraints

1. The microgrid electric quantity balance constraint is:

$$P_{WT,\tau}^{DA} \times \frac{24}{\omega} + P_{PV,\tau}^{DA} \times \frac{24}{\omega} + \sum_{k}^{n} \sum_{s=1}^{S} Q_{c,k,s,\tau}^{DA} + Q_{b,\tau}^{DA} + Q_{r,\tau}^{DA} = Q_{on,\tau}^{DA} + Q_\tau^{DA\prime} \tag{20}$$

where $P_{WT,\tau}^{DA}$ and $P_{PV,\tau}^{DA}$ are the day-ahead predicted power of WT and PV units in the time period of $\tau$, respectively; and $\frac{24}{\omega}$ is the unit time period of the spot market, which is used to approximately convert power into electricity.

Due to factors such as the stochastic output of WT and PV and the uncertainty of the spot price, the microgrid cannot guarantee that the declared electricity quantity is completely consistent with the required electricity quantity in the day-ahead forecast. However, a certain confidence level must be satisfied. Therefore, the power balance constraint can be converted into a chance constraint:

$$C_r\left\{ -\sigma \leq P_{WT,\tau}^{DA} \times \frac{24}{\omega} + P_{PV,\tau}^{DA} \times \frac{24}{\omega} + Q_{b,\tau}^{DA} + \sum_{k}^{n} \sum_{s=1}^{S} Q_{c,k,s,\tau}^{DA} + Q_{r,\tau}^{DA} - Q_{on,\tau}^{DA} - Q_\tau^{DA\prime} \leq \sigma \right\} \geq \beta \tag{21}$$

where $\sigma$ is the allowable deviation of unbalanced power.

2. The microgrid interactive power constraint (Internet access and power purchase cannot be carried out simultaneously) is:

$$Q_{on,\tau}^{DA} \times Q_{r,\tau}^{DA} = 0 \tag{22}$$

3. The stochastic micro-source output constraint is:

$$0 \leq P_{WT,\tau}^{DA} \leq P_{WT}^{R} \tag{23}$$

$$0 \leq P_{PV,\tau}^{DA} \leq P_{PV}^{R} \tag{24}$$

where $P_{WT}^{R}$ and $P_{PV}^{R}$ are the rated power of the WT and PV unit, respectively.

4.  The controllable micro-source output constraint are Equations (6) and (7).
5.  The battery energy storage state and charge/discharge rate constraint are Equations (8) and (9).
6.  The demand response power balance constraint is:

$$\sum_{\tau=1}^{\omega} Q_{\tau}^{DA\prime} = \sum_{\tau=1}^{\omega} Q_{\tau}^{DA} \tag{25}$$

### 3.3. The Real-Time Market Microgrid Scheduling Model

In the real-time market stage, penalty fees will be charged for the deviation between the actual purchased electricity in the real-time market and the declared electricity in the day-ahead market in order to encourage users to reasonably declare the day-ahead demand curve and prevent them from making profits by exploiting loopholes in the market rules. Therefore, in the process of participating in the real-time market, microgrids should consider the penalty fee of deviation power in addition to forecasting the electricity prices in order to achieve optimal scheduling. The optimal scheduling model (where the real-time electricity price forecasting methods and the conventional constraints are consistent with the day-ahead scheduling, and will not be described again) can be expressed as:

$$\begin{cases} \max \overline{f_{RT}} \\ C_r \left\{ f_{RT} \geq \overline{f_{RT}} \right\} \geq \alpha \\ C_r \left\{ -\sigma \leq P_{WT,\tau}^{RT} \times \frac{24}{\omega} + P_{PV,\tau}^{RT} \times \frac{24}{\omega} + Q_{b,\tau}^{RT} + \sum_{k}^{n} \sum_{s=1}^{S} Q_{c,k,s,\tau}^{RT} + Q_{a,\tau}^{RT} - Q_{on,\tau}^{RT} - Q_{\tau}^{RT\prime} \leq \sigma \right\} \geq \beta \end{cases} \tag{26}$$

$$f_{RT} = R_s^{RT} - C_c^{RT} - C_b^{RT} - C_d^{RT} - C_p^{RT} \tag{27}$$

$$R_s^{RT} = \sum_{\tau=1}^{\omega} P_{s,\tau} Q_{\tau}^{RT\prime} - P_{c,\tau}^{DA} Q_{r,\tau}^{DA} + P_{on} Q_{on,\tau}^{RT} + \left( Q_{r,\tau}^{DA} - Q_{a,\tau}^{RT} \right) P_{c,\tau}^{RT} \tag{28}$$

$$C_p^{RT} = \left| Q_{r,\tau}^{DA} - Q_{a,\tau}^{RT} \right| \times P_p \tag{29}$$

where $R_s^{RT}$ is the real-time market income based on the day-ahead market's income; $C_p^{RT}$ is the deviation electricity penalty fee; $Q_{\tau}^{RT\prime}$ is the actual power consumption after the user participates in the demand response in the time period of $\tau$; $Q_{on,\tau}^{RT}$ is the on-grid energy of the microgrid in the time period of $\tau$; $Q_{a,\tau}^{RT}$ is the actual power purchased by the microgrid in the time period of $\tau$; $P_{c,\tau}^{RT}$ is the real-time market's electricity price in the time period of $\tau$; and $P_p$ is the unit electricity penalty fee.

## 4. The Solving of Model

In the first stage, two kinds of uncertain factors (i.e., clearing price and output of WT in the day-ahead market) are dealt with by using the prediction model and stochastic simulation method. Then, the PSO algorithm was used to find the optimal solution of each unit's output and the declared electricity quantity in the day-ahead market, and the result was input to the next stage.

The second stage uses the same method to deal with the real-time market price and the output of the random power supply of the microgrid. Taking the optimization result of the first stage and the day-ahead market clearing price as the known quantity, we introduced the penalty cost of the electricity quantity's deviation. Aimed at the maximum revenue, the PSO was used to obtain the optimal scheduling results of the microgrid in the real-time market. The specific solution methods of the first stage or the second stage are as follows:

### 4.1. Stochastic Simulation

In view of the existence of chance constraints in the established model, stochastic simulation technology was introduced to process variables, and the steps are as follows:

(1) Solve the objective function. (1) Randomly generate mutually independent variables $\varepsilon_k$ ($k = 1, 2, \ldots, n$), according to the probability distribution of random variables in the objective function; (2) calculate a function value $f_k = f(x, \varepsilon_k)$; and (3) set $\bar{n} = \alpha n$, and the $\bar{n}$ th maximum in $\{f_1, f_2, \ldots, f_n\}$ is the target value according to the law of large numbers.
(2) Test the chance constraint. (1) Set $\bar{n} = 0$; (2) randomly generate a variable $\varepsilon$, according to the probability distribution of random variables in constraint conditions; (3) calculate $g(x, \varepsilon)$, and $\bar{n} = \bar{n} + 1$, if it is less than or equal to 0; (4) after repeating steps (2) and (3) for $n$ times, the constraint holds if $\bar{n}/n \geq \beta$.

### 4.2. Particle Swarm Optimization Algorithm

After dealing with random variables, the PSO based on the Monte Carlo simulation was used to solve the model. The specific process is as follows:

(1) Initialize data input. Input the deterministic values such as the controllable micro-source, battery cost parameters and load demand as well as confidence level and algorithm control parameters.
(2) Population initialization. With the output of WT and PV and the spot electricity price as random variables, random variables are generated from probability distribution to form an initial population in which all particles satisfy the constraint under the constraint condition test and by using the above random simulation technology, and the population position and velocity are initialized.
(3) The population target value is obtained as the fitness, and the individual extremum and global extremum are judged based on the fitness.
(4) Update the particle position and velocity. Iterative calculation is carried out to generate a new generation of particle swarms. The position and velocity are updated as follows:

$$v_\rho^{i+1} = \Psi v_\rho^i + l_1 r_1 \left( P_{\rho OT}^i - L_\rho^i \right) + l_2 r_2 \left( g_{\rho OT}^i - L_\rho^i \right) \tag{30}$$

$$L_\rho^{i+1} = L_\rho^i + v_\rho^{i+1} \tag{31}$$

where $\Psi$ is the inertia weight; $v_\rho^i$ and $L_\rho^i$ are the velocity and position of the *i*th iteration of particle $\rho$, respectively; $l_1$ and $l_2$ are learning factors; $r_1$ and $r_2$ are random numbers between [0, 1]; and $P_{\rho OT}^i$ and $g_{\rho OT}^i$ are the individual extremum of the *i*th iteration of particle $\rho$ and the global extremum of the population, respectively.
(5) Repeat steps (2)–(4) to search for the optimal solution until the maximum iteration numb is reached. Output that optimal solution and the corresponding particle position.

Compared with deterministic programming problem solving, the solution method used in this paper first deals with the uncertain factors in the model. On this basis, the PSO algorithm, which can find the optimal solution in a certain range, was used to solve the model. The flow chart (Figure 3) of the PSO algorithm combined with stochastic simulation technology can be shown as follows.

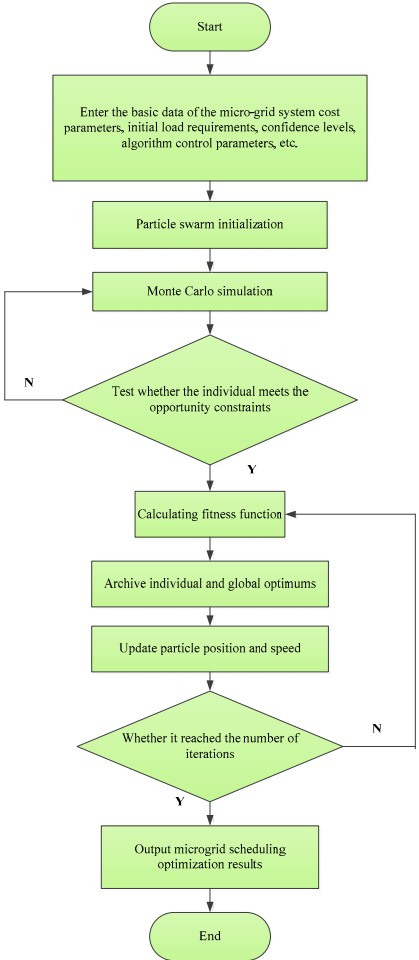

**Figure 3.** The flow chart of the algorithm solution.

*4.3. Convergence Test*

In this paper, the PSO was improved by stochastic simulation technology to solve the two-stage optimal scheduling model of a microgrid based on chance-constrained programming. The convergence curve of the PSO can be shown in Figure 4.

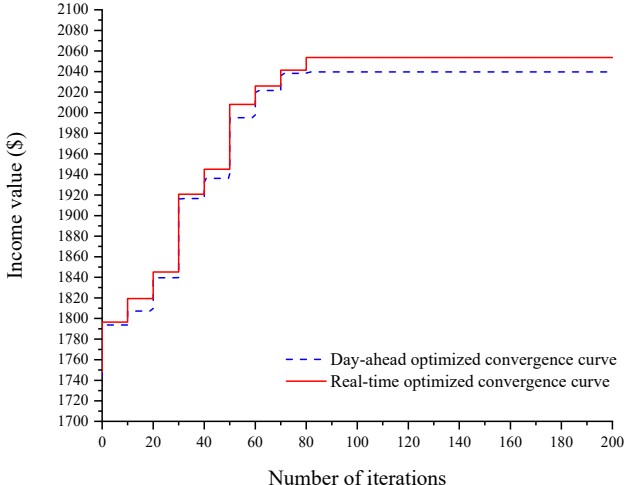

**Figure 4.** The scheduling revenue convergence curve of the microgrid.

## 5. Results and Discussion

### 5.1. Basic Data

A microgrid that integrated with WT, PV, DG, and ESS, was selected to simulate its participation in the spot market trading process to verify the rationality of the proposed model. The total output power of the DG in the microgrid was 1 MW, and the operation cost parameters of units are shown in Table 1. Based on the power curve of a typical load day, the user load was converted into electricity consumption in each period, as shown in Table 2. The other parameters used in the empirical analysis are shown in Table 3.

**Table 1.** The operation cost parameter of diesel generator

| Unit | Output Power (kW) | | Sectional Rated Power (kW) | | | Segmental Average Cost ($/kWh) | | |
|---|---|---|---|---|---|---|---|---|
| | Lower Limit | Upper Limit | The First Segment | The Second Segment | The Third Segment | The First Segment | The Second Segment | The Third Segment |
| 1 | 0 | 200 | 100 | 150 | 200 | 433.71 | 490.59 | 540.36 |
| 2 | 0 | 300 | 125 | 275 | 300 | 604.35 | 668.34 | 753.66 |
| 3 | 0 | 500 | 200 | 350 | 500 | 959.85 | 1137.6 | 1279.8 |

**Table 2.** The electricity data of a typical load day.

| Period | Electricity/kWh | Period | Electricity/kWh | Period | Electricity/kWh |
|---|---|---|---|---|---|
| 1 | 1031.94 | 9 | 1760.01 | 17 | 1706.92 |
| 2 | 963.69 | 10 | 1833.95 | 18 | 1739.15 |
| 3 | 912.50 | 11 | 1832.06 | 19 | 1727.78 |
| 4 | 878.37 | 12 | 1801.72 | 20 | 1596.95 |
| 5 | 834.76 | 13 | 1608.33 | 21 | 1466.13 |
| 6 | 891.64 | 14 | 1488.88 | 22 | 1107.78 |
| 7 | 1105.89 | 15 | 1511.63 | 23 | 1079.34 |
| 8 | 1367.54 | 16 | 1557.14 | 24 | 1058.49 |

**Table 3.** Relevant parameters.

| | | |
|---|---|---|
| WT | rated total power | 3 MW |
| | cut in wind speed | 4 m/s |
| | cut off wind speed | 25 m/s |
| | rated wind speed | 16 m/s |
| | shape parameters of Weibull Distribution | 2.2 |
| PV | total rated power | 8 MW |
| | total area of PV modules | 1.2 hm$^2$ |
| | efficiency of photoelectric conversion | 24% |
| | shape parameters of Beta Distribution | 0.5 |
| ESS | recommended capacity | 1.5 MWh |
| | charging efficiency | 0.9 |
| | discharge efficiency | 0.9 |
| Electricity price | sales price | 0.0851 $/kWh |
| | feed-in price | 0.0589 $/kWh |
| | penalty for deviation | 0.0356 $/kWh |

It was assumed that all purchased power generated by the microgrid in the spot transaction process was all from the spot market. The actual data of a certain day in the Pennsylvania—New Jersey—Maryland (PJM) electric power market were used to forecast the day-ahead clearing price and the real-time price, and $\varpi$ was taken as 24. By using the autoregressive model in Section 3.2.1,

we could obtain the day-ahead market price and real-time market price, which are shown in Figure 5. Considering that the granularity of the day-ahead market clearing in China is 15 min, it was assumed that the forecast electricity price is the arithmetic average of four 15-min clearing rates per hour.

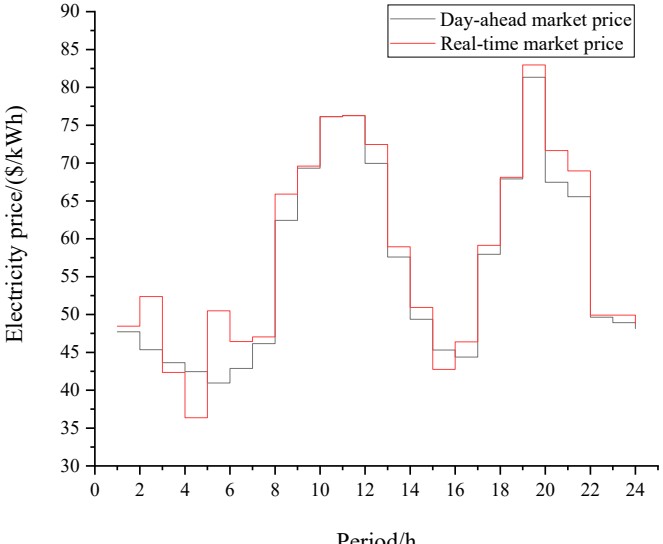

**Figure 5.** Spot market electricity price forecast chart.

## 5.2. Result Analysis

### 5.2.1. Scheduling Operation Analysis

We set the confidence level of the objective function and the constrain condition as 0.9. While the particle swarm size was 80, the learning factor was 2, and the inertia weight was 0.8. The maximum number of iterations was 200. First, the demand response model was eliminated to optimize the microgrid, and the first-stage scheduling model was run for a comparison with the forecast electricity price of the day-ahead market (Figure 6, ME is the interactive power between the microgrid and main network).

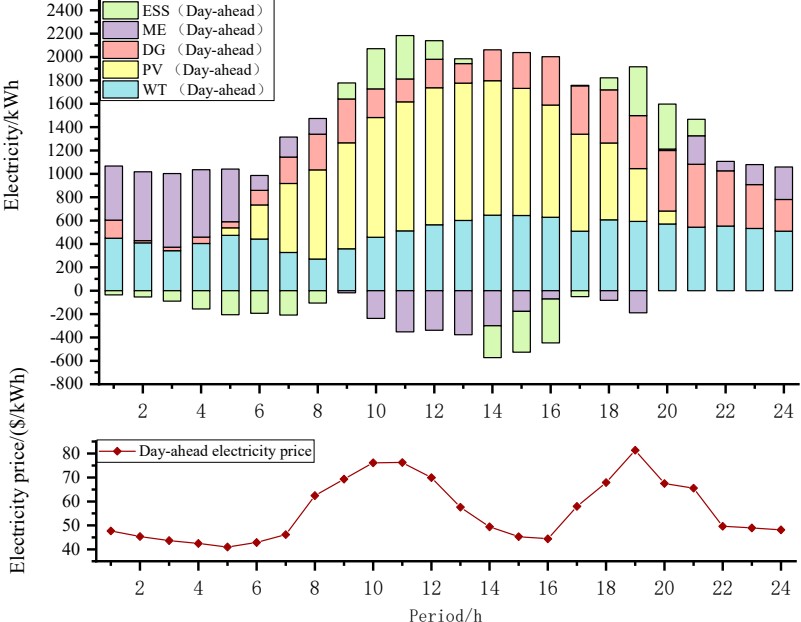

**Figure 6.** The comparison of the microgrid operation plan and electricity price in the day-ahead market.

As can be seen from Figure 6, during the period when the forecast electricity price level is low, the microgrid tends to declare more electricity to meet its own electricity demand and continuously charge the ESS. With the increase in electricity price, the DG output increases and gradually replaces the spot market electricity to fill the shortage of the new energy electricity supply capacity. When operating to a higher electricity price level, the ESS starts to discharge and provides backup assistance jointly with DG. At this time, the microgrid will basically no longer purchase electricity from the spot market, but will instead put the remaining electricity on the grid. During the full load day, the total electricity declared in the day-ahead stage was 3934.21 kWh, which accounted for 11.97% of the forecast power on that day. The maximum electricity of DG per unit period was 392.56 kW. Compared with the total output electricity of 1 MW, there was still a large margin to bear the random fluctuation of the spot market electricity price. It can be seen that through the first stage of market optimization scheduling, the microgrid has high reliability while being economically optimal.

By continuing to run the second-stage scheduling model, the calculation results are shown in Figure 7. Additionally, a comparison of the two-stage scheduling results can be obtained, which is shown in Figure 8.

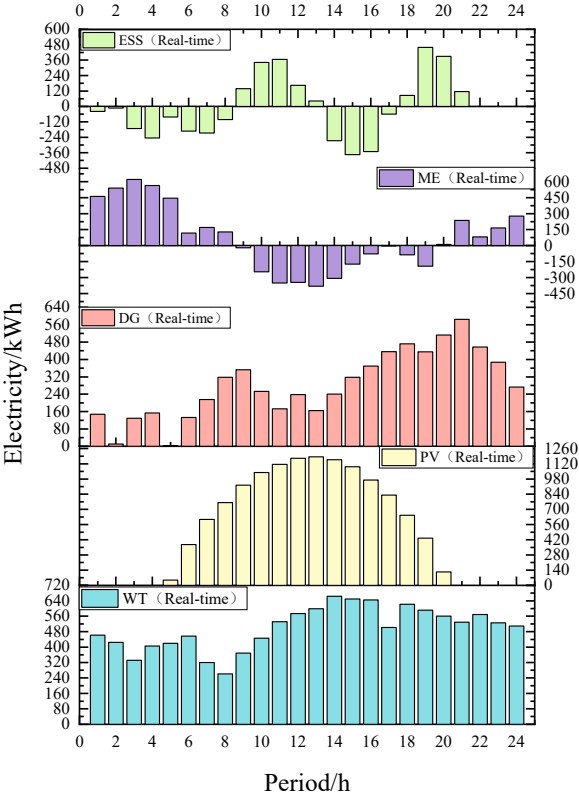

**Figure 7.** Real-time market microgrid operation plan.

As can be seen from Figures 7 and 8, in the real-time market scheduling stage, the operation plans of ESS and DG experienced severe fluctuations in some time periods, while the planned purchase curve of the spot market (two stage power purchase decision optimization curve, i.e., the part of the ME curve > 0 in Figure 8) fluctuated slightly. Its maximum fluctuating power per unit period was only 8.63 kWh. This is because in the second stage, when the predicted electricity price and the stochastic micro-source predict the output change, the microgrid comprehensively considers the operation cost of each unit and the deviation power penalty fee and chooses to actively adjust the ESS and DG unit with relatively small variable costs to adapt to the changes of uncertain factors and realize the maximum economic benefit of microgrid.

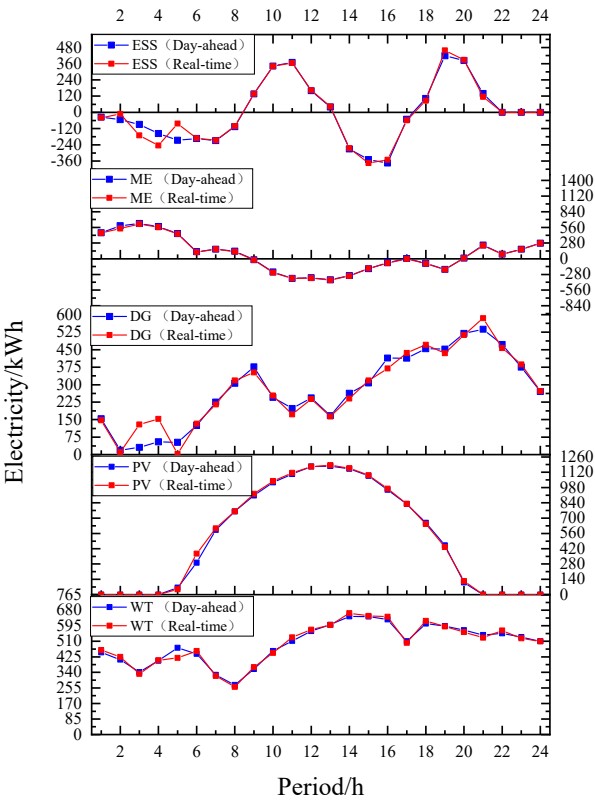

**Figure 8.** The comparison of the two-stage optimal scheduling results.

5.2.2. The Impact Analysis of Demand Side Management

In order to further improve the optimal allocation of power resources, the demand side management was introduced to solve the model with the advantage that microgrid users can easily adjust load. Suppose that the microgrid uses the time-of-use electricity price to guide users to transfer load. The electricity price elasticity matrix adopts the elasticity matrix of large industrial users in this paper [24–26]. With the current sales price as the normal period electricity price, $\gamma$ is defined as the floating proportion of the sales price. While the peak electricity price rises by $1 + \gamma$ on this basis, the valley electricity price is lowered by $1 - 1.2\gamma$. The time division is shown in Table 4. Let $\gamma$ float between $[0, 0.8]$ in each step of 0.05 to obtain the microgrid real-time market income change as shown in Table 5.

**Table 4.** The period division of the time-of-use price.

|  | **Flat** | **Peak** | **Valley** |
|---|---|---|---|
| time division (h) | 12:00–17:00 21:00–24:00 | 8:00–12:00 17:00–21:00 | 0:00–8:00 |
| price standard ($/kWh) | 0.0851 | $0.0851 \times (1 + \gamma)$ | $0.0851 \times (1 - 1.2\gamma)$ |

The results in Table 4 show that when the microgrid participates in spot market transactions, the implementation of demand side management can significantly improve its own economic benefits. With the increase of peak-to-valley price difference, the income shows a trend of increasing first and then decreasing. When the floating ratio was 0.55, the profit reached its highest. By analyzing the cost of each factor under different electricity prices, it can be found that after the implementation of time-of-use electricity price, the user reduces the electricity bill expenditure by transferring more flexible loads. The load transfer also promotes the coupling of the power consumption curve and renewable energy output, changes the co-directional distribution of the microgrid power declaration curve and the spot market electricity price, then reduces the reserve cost of the microgrid during

peak hours and reduces the average spending on purchasing electricity. However, with the increasing peak-to-valley electricity price ratio, the sensitivity of users to electricity price decreases until it has been "saturated". Since then, the benefit increment of the microgrid due to the user demand response starts to be lower than the electricity bills saved by users, and the revenue declines.

**Table 5.** The income change of the microgrid under different floating ratios in the real-time market.

| Initial Sales Price ($/kWh) | | Before the Demand Response ($) | |
|---|---|---|---|
| 0.0851 | | 1998.76 | |
| Floating Rate (%) | Income ($) | Floating Rate (%) | Income ($) |
| 5% | 2018.81 | 45% | 2172.53 |
| 10% | 2042.70 | 50% | 2188.17 |
| 15% | 2064.18 | 55% | 2192.72 |
| 20% | 2088.63 | 60% | 2190.16 |
| 25% | 2105.56 | 65% | 2182.20 |
| 30% | 2122.90 | 70% | 2173.95 |
| 35% | 2139.11 | 75% | 2165.85 |
| 40% | 2155.61 | 80% | 2157.60 |

By setting the electricity price floating ratio to 0.55, the comparison of the day-ahead scheduling results before and after demand side management implementation of the microgrid, and the comparison of the two-stage scheduling results after the implementation of demand side management can be shown in Figure 9, Table 6 and Figure 10, respectively.

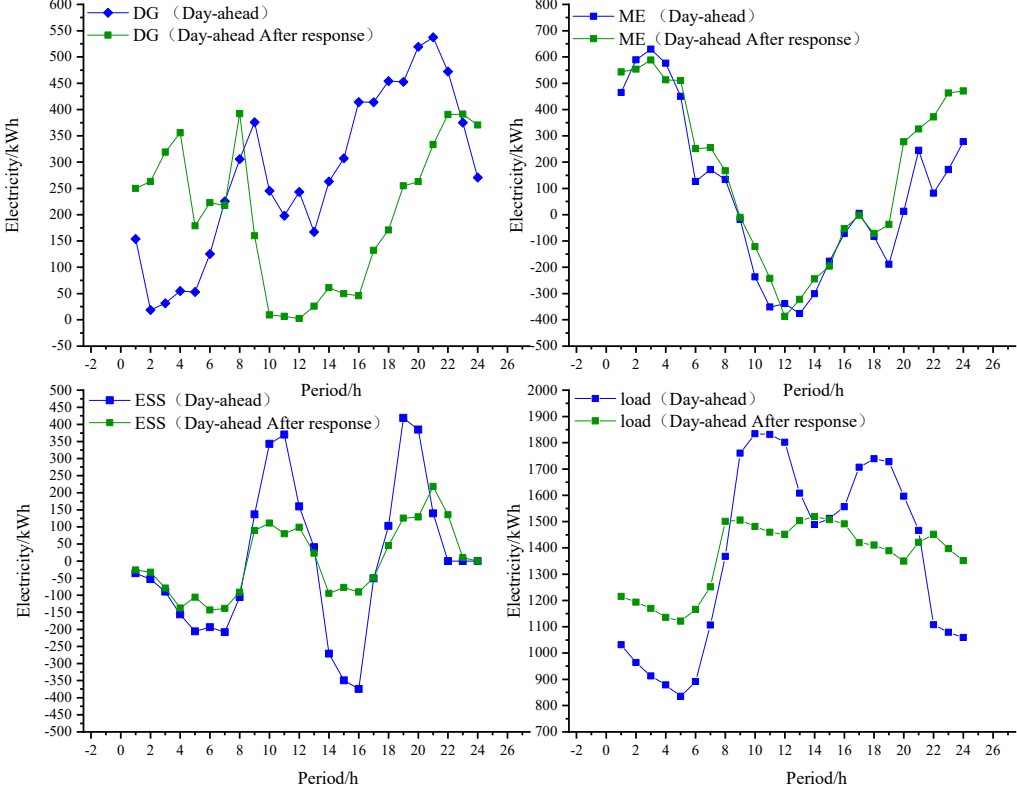

**Figure 9.** The comparison of day-ahead scheduling of the microgrid before and after demand side management.

**Table 6.** Comparison of the key parameters before and after the implementation of demand response.

|  | Max (load) | Min (load) | Max (ME) | Max (DE) | Max (ESS) | Max (ESS) |
|---|---|---|---|---|---|---|
| Before demand response | 1835.95 | 833.76 | 635.32 | 547.41 | 422.79 | 378.69 |
| After demand response | 1516.44 | 1123.44 | 582.89 | 387.56 | 218.55 | 139.35 |

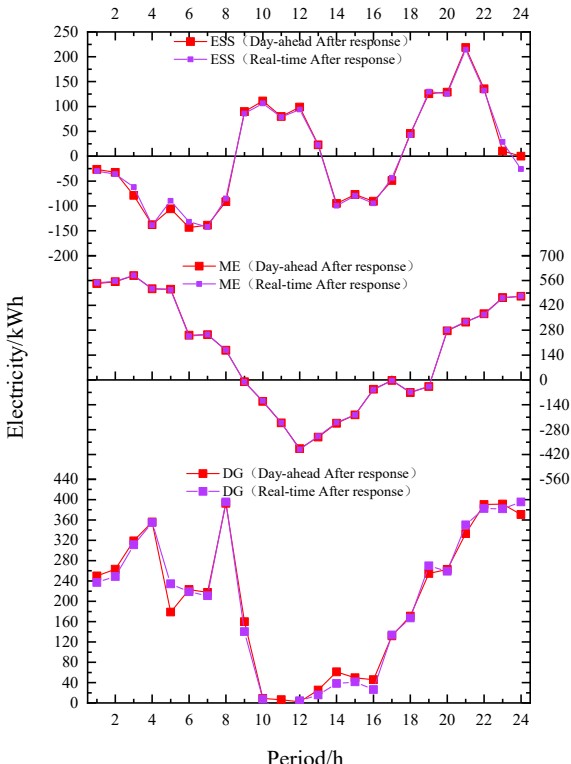

**Figure 10.** The comparison of the two-stage optimal scheduling results (demand side management).

As shown in Figure 9 and Table 6, after demand side management is implemented in the microgrid, the user load curve and the microgrid scheduling unit output curve both tend to be flat, compared with before. The specific changes are as follows:

The maximum electricity consumption per unit period of the user was reduced by 319.51 kWh, the minimum electricity consumption increased to 289.68 kWh, and the maximum declared electricity per unit period was reduced from 635.32 kWh to 582.89 kWh. While the maximum output power of the DG decreased from 547.41 kW to 387.56 kW, the maximum charging capacity of the ESS per unit period decreased from 422.79 kWh to 218.55 kWh, and the maximum discharge decreased from 378.69 kWh to 139.35 kWh.

This is because under the stimulation of demand side management, the users in the microgrid can reduce the maximum electricity consumption and increase the minimum electricity consumption by changing the electricity demand in the peak and valley period to realize the peak load shifting. The change in load demand makes the output of DG and charge and discharge behavior of EES change correspondingly.

Additionally, by comparing Figure 10 with Figure 8, the adjustments in the real-time scheduling phase of the microgrid have decreased compared with the previous ones, and the ability of the microgrid to participate in the spot market to respond to changes in uncertain factors has increased after the implementation of demand side management.

### 5.2.3. Confidence Level Impact Analysis

The influence of confidence level on the microgrid's two-stage optimal scheduling was analyzed. Set $\alpha$ and $\beta$ change between 0.8–1, respectively, and run the microgrid scheduling model and real-time market microgrid scheduling model in Section 3, respectively. Under different confidence combinations, we can obtain the microgrid revenue curve of the day-ahead market and real-time market, as shown in Figures 11 and 12. It can be seen from Figures 11 and 12 that when the confidence level of objective function and constraint condition is low (the red part in the figure), the economic benefit of the microgrid is better. With the continuous increase of the objective function and constraint condition's confidence level, the economic benefit shows a downward trend. When the confidence level is high (the purple part in the figure), the economic benefit of the microgrid is worse. With the improvement of the confidence level, the requirements for the establishment of the objective function and constraints expressed in probability form are gradually strict, and the power balance constraint range is reduced. It is not conducive to the calculation of better optimization results, but conducive to improving the reliability of microgrid operation.

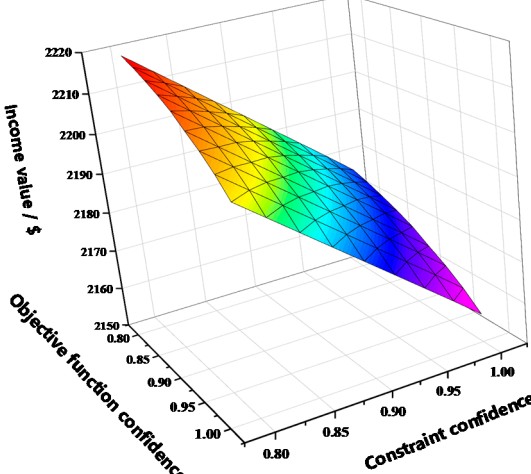

**Figure 11.** The curve diagram of the microgrid's day-ahead market returns under different levels.

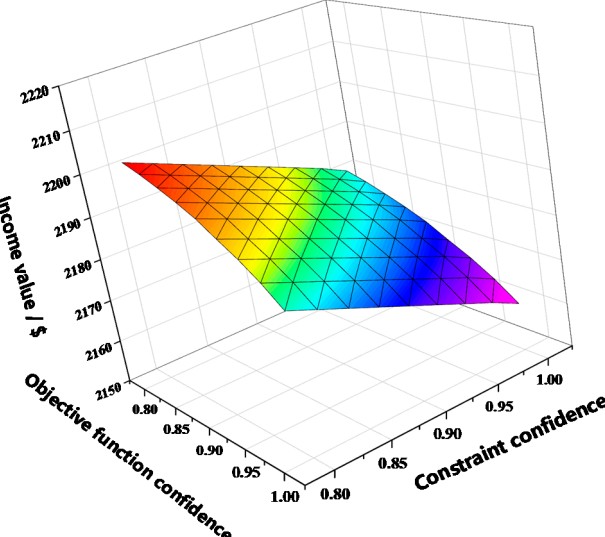

**Figure 12.** The curve diagram of the microgrid's real-time market returns under different levels.

In addition, compared with Figures 11 and 12, it can be seen that with the improvement in confidence level, the economic benefit of the microgrid's day-ahead market scheduling declines faster

than real-time market scheduling's. This is because there are relatively more uncertainty factors faced by the day-ahead market, and the economic cost of improving microgrid reliability is greater than that of the real-time market.

Therefore, in the two-stage scheduling optimization process of the microgrid, the improvement in the confidence level will make the test of objective function and constraints stricter, enhancing the reliability of microgrid operation. At the same time, however, it will lead to a certain degree of decline in the economic benefits of the microgrid, and the economic benefits decline faster in the current market stage due to greater uncertainty. Therefore, the decision maker can choose the confidence level considering the risk aversion degree, and adjust the balance between the microgrid economy and reliability, thus achieving optimal scheduling.

## 6. Conclusions and Policy Implications

According to the basic structure of a microgrid, a microgrid system model considering the stochastic output of WT and PV was established in this paper. On this basis, considering various uncertain factors comprehensively and with the aim of maximizing the benefits of a microgrid system at a certain confidence level, a two-stage scheduling model based on chance-constrained programming was put forward, combined with the characteristics of the spot market. The purpose of this paper was to make a two-stage optimal scheduling plan for microgrids participating in the electricity spot market by optimizing the power supply side and load side simultaneously, and improving the economic benefits of the microgrid system. The following conclusions can be drawn:

(1) The microgrid can effectively avoid potential risks by using the model proposed in this paper to make spot market decisions. Through two-stage optimal scheduling, it can deal with the fluctuation of WT and PV output and electricity price by adjusting system resources, which can both achieve higher reliability and pursue maximum benefits.

(2) The demand side management should be introduced at the time of optimizing the supply-side resources. Thus, the two-stage economic benefits of the microgrid can be improved, the reserve capacity and declared power during the peak period of day-ahead market can be lowered, the output adjustment of the real-time market to each unit can be reduced, and then the microgrid's ability to resist risks can be significantly improved.

(3) The confidence level will have a certain impact on the results of the two-stage scheduling of the microgrid. While risk-prone people running the model with a low confidence level can obtain better economic benefits, risk-averse people running the model with a high confidence level can be conducive to ensuring the safety and stability of the microgrid.

**Author Contributions:** J.L.'s contribution is formal analysis and writing—review & editing; C.T.'s contribution is data curation and formal analysis; Z.R.'s contribution is data curation and resources; J.Y.'s contribution data curation and methodology; X.Y.'s contribution is data curation; Z.T.'s contribution is formal analysis and supervision. All authors have read and agreed to the published version of the manuscript.

**Funding:** The work was supported by the National Science Foundation of China (Grant No. 71573084), and the 2018 Key Projects of Philosophy and Social Sciences Research, Ministry of Education, China (Grant No. 18JZD032).

**Acknowledgments:** The completion of this paper has been helped by many teachers and classmates. We would like to express our gratitude to them for their help and guidance.

**Conflicts of Interest:** The authors declare no conflict of interest.

**Nomenclature**

Acronyms

| | |
|---|---|
| WT | Wind turbine |
| PV | Photovoltaic |
| ESS | Energy storage system |
| DG | Diesel generator |
| ME | Mutual energy |
| PSO | Particle swarm optimization |

Variables

| | |
|---|---|
| $f_{WT}(v)$ | Probability density function of actual wind speed |
| $P_{WT}$ | Output power of WT (kW) |
| $f_{PV}(l)$ | Probability density function of light intensity |
| $P_{WT}$ | Output power of PV unit (kW) |
| $C_c$ | Total operation cost of controllable unit ($) |
| $SOC_\tau$ | The charged state of battery energy storage in $\tau$ period |
| $C_b$ | Depreciation loss cost caused by battery charging and discharging ($) |
| $C_d$ | Demand side management cost ($) |
| $P_{c,\tau}^{DA}$ | Market clearing electricity prices in the time period of $\tau$ in the day-ahead market in the current day ($/kWh) |
| $R_s^{DA}$ | Market clearing income of the microgrid day-ahead market ($) |
| $P_{s,\tau}$ | Users' electricity price in the time period of $\tau$ ($/kWh) |
| $Q_{r,\tau}^{DA}$ | Microgrid day-ahead declared power in the time period of $\tau$ (kWh) |
| $Q_\tau^{DA\prime}$ | Predicted response electricity in the time period of $\tau$ (kWh) |
| $P_{on}$ | Microgrid on-grid electricity price ($/kWh) |
| $Q_{on,\tau}^{DA}$ | Predicted on-grid power in the time period of $\tau$ (kWh) |
| $P_{WT,\tau}^{DA}$ | Day-ahead predicted power of WT in the time period of $\tau$ (kW) |
| $P_{PV,\tau}^{DA}$ | Day-ahead predicted power of PV units in the time period of $\tau$ (kW) |
| $R_s^{RT}$ | Real-time market income based on the day-ahead market's income ($) |
| $C_p^{RT}$ | Deviation electricity penalty fee ($) |
| $Q_\tau^{RT\prime}$ | Actual power consumption after the user participates in the demand response in the time period of $\tau$ (kWh) |
| $Q_{on,\tau}^{RT}$ | On-grid energy of the microgrid in the time period of $\tau$ (kWh) |
| $Q_{a,\tau}^{RT}$ | Actual power purchased by the microgrid in the time period of $\tau$ (kWh) |
| $Q_{c,\tau}^{RT}$ | Actual power purchased by the microgrid in the time period of $\tau$ (kWh) |
| $P_{c,\tau}^{RT}$ | Real-time market's electricity price in the time period of $\tau$ ($/kWh) |

Parameters

| | |
|---|---|
| $k$ | Shape parameters of Weibull Distribution |
| $c$ | Scale parameters Weibull Distribution |
| $v_{ci}$ | Cut in wind speed (m/s) |
| $v_{co}$ | Cut off wind speed (m/s) |
| $p_r$ | Rated power (MW) |
| $v_r$ | Rated wind speed (m/s) |
| $a_l, b_l$ | Shape parameters of Beta Distribution |
| $l_{max}$ | Maximum solar irradiance |
| $S_{PV}$ | Total area of PV modules (hm$^2$) |
| $\eta$ | Efficiency of photoelectric conversion |
| $\omega$ | Number of spot market periods |
| $S$ | Total number of segments of linear operation cost of controllable unit |
| $N$ | Total number of controllable units |
| $C_{c,n,s}^A$ | Average generation cost of the nth unit in section $s$ ($) |
| $Q_{c,n,s,\tau}$ | Generation capacity of unit $n$ in section $s$ in period $\tau$ (kWh) |
| $Q_{c,n,s}^R$ | Rated generating capacity of unit $n$ in section $s$ (kWh) |
| $Q_{b,\tau}$ | Charge and discharge capacity of ESS in $\tau$ period (kWh) |

| | |
|---|---|
| $\eta_{in}, \eta_{out}$ | Charge and discharge efficiency respectively |
| $W_b$ | Battery capacity (MWh) |
| $SOC_{min}$ | Lower limit of charged state |
| $SOC_{max}$ | Upper limit of charged state |
| $u_b$ | Battery loss factor |
| $Q_\tau^0$ | User power consumption in the $\tau$ period before demand response (kWh) |
| $Q_\tau'$ | User power consumption in the $\tau$ period after demand response (kWh) |
| $P_\tau^0$ | User electricity price in the $\tau$ period before demand response ($/kWh) |
| $P_\tau'$ | User electricity price in the $\tau$ period after demand response ($/kWh) |
| $P_\varsigma^0$ | User electricity price in the $\varsigma$ period before demand response ($/kWh) |
| $P_\varsigma'$ | User electricity price in the $\varsigma$ period after demand response ($/kWh) |
| $e_{\tau\tau}$ | Self-elasticity |
| $e_{\varsigma\tau}$ | Cross elasticity |
| $\Delta Q_\tau$ | Change of user power consumption in the $\tau$ period (kWh) |
| $\Delta P_\tau$ | Change of electricity price of users in the $\tau$ period ($/kWh) |
| $C(\tau)$ | Average value of market clearing electricity price in the time period of $\tau$ in the day-ahead market ($/kWh) |
| $\phi_1$ | Autoregressive coefficient |
| $\varepsilon_\tau$ | White noise sequence |
| $\sigma$ | Allowable deviation of unbalanced power |
| $P_{WT}^R$ | Rated power of WT (MW) |
| $P_{PV}^R$ | Rated power of PV unit (MW) |
| $P_p$ | Unit electricity penalty fee ($/kWh) |
| $\psi$ | Inertia weight |
| $v_\rho^i$ | Velocity of the $i$th iteration of particle $\rho$ |
| $L_\rho^i$ | Position of the $i$th iteration of particle $\rho$ |
| $l_1, l_2$ | Learning factors |
| $r_1, r_2$ | Random numbers between $[0, 1]$ |
| $P_{\rho OT}^i$ | Individual extremum of the $i$th iteration of particle $\rho$ |
| $g_{\rho OT}^i$ | Global extremum of the population |

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
