# Peer review of "A Two-Stage Optimal Scheduling Model of Microgrid Based on Chance-Constrained Programming in Spot Markets"

_processes, doi:10.3390/pr8010107_

Round 1

Reviewer 1 Report

The manuscript deals with the optimal scheduling model of microgrid in spot markets. 

The topic of spot markets and various methods based on optimizations or programming algorithm are extensively presented in literature. The main weakness of this paper is the literature review. I think the authors should deepen this aspect in order to highlight the novelty of their work. In fact, on my opinion, it is not clear what added value the paper should provide with respect to the existing literature. 

Nonetheless, the microgrid structure is interesting and this should be given more emphasis. 

With respect to the model, I think the approach can be to some extent considered valid, but it is really disorganized. Authors should identify a leitmotiv and follow it. For example, it is not sufficiently clear how the calculations in paragraoh 2 support the solving of the model.

Similar discussion for the results. They are not clear exposed and the paper is difficult to read and undestand, at least for what the aim is concerned.

Reviewer 2 Report

The paper is generally interesting and well-drafted, but some important issues occur, which should be addressed properly, as detailed in the following:

- a nomenclature section should be enclosed at the start of the paper, which summarizes all the symbols employed throughout the paper, together with their meaning and unit. In this regard, mathematical modelling should be revised carefully, by avoiding, when possible, the use of very complex symbols characterized by many superscripts and/or subscripts;

- please increase the graphical resolution of all the figures reported in the paper as many of them now show unreadable values (especially Figs. 8-9);

-considering (5) and lines 127-131, it seems that both τ and s are expressed in hours: please clarify this aspect, also by defining these indexes in the nomenclature section. In addition, k in the sum of (5) should be replaced with “k = 1”;

- considering (8), ηout should appear in the denominator as Qb,τ is positive;

- Section 3 is difficult to read and understand, especially due to the complex mathematical symbols employed. In this regard, I suggest clarifying the meaning of each equation extensively by words and simplifying the mathematical symbols and expressions to the maximum extent. Moreover, one or more flowcharts and tables should be enclosed in order to clarify how the proposed two-stage procedure works: for example, a table can summarize all the constraints considered by each stage of the procedure, together with the optimization targets, while a flowchart could explain how the two stages work and their connection to each other;

- Section 3, (17)-(18) are unclear, please explain the meaning of each term of those equations. The same applies also to (19), (21) and to the term 24/ω, which is defined as a time period, but which seems a ratio between hours and, thus, a dimensionless coefficient;

- Section 4 seems describing the mathematical solver used for computing the optimal solution for both stages: if so, I suggest enclosing a flowchart for better describing the solving method and moving Fig. 10 to this section (lines 407-412). Possibly, this section should be converted into a subsection of Section 3, especially if it will be quite short;

- Section 5, please report all the numerical values in dedicated tables rather than within the main text (lines 282-304). In this regard, please specify which is the ESD charging and discharging efficiency precisely (about 90% is not acceptable);

- Section 5, please better clarify how day-ahead and real-time market prices shown in Fig. 2 are computed;

- Section 5, lines 324-325, it seems that “power” should be replaced by “electricity” or “energy” as it is given in kWh. In addition, the concept of “electricity/power declared on the day-ahead market” should be explained;

- Section 5, lines 338-339, the planned purchase curve to which these lines refer does not appear neither in Fig. 4 nor in Fig. 5, please clarify. In addition, please quantify and highlight the benefits of real-time market scheduling compared to using just the day-ahead scheduling, which seems marginal, especially in the case of using demand-response program (Fig. 7);

- Section 5, lines 381-391, please summarizes electricity values there reported in a table and explain their meaning.

In addition, the following minor issues should be addressed:

- I suggest using Energy Storage System (ESS) and Diesel Generator (DG) instead of Energy Storage Device (ESD) and Diesel Engine (DE) throughout the paper;

- considering (1), the square brackets should be removed (they do not seem necessary) and f should be replaced by “fWT” in order to avoid misunderstanding with (3) and some following equations (for example, (15));

- Section 2.1, please replace “fan” with “turbine” or “WT”;

- considering (3), please define Γ and replace f with “fPV”;

- Section 2.2, line 120, please add “generators” after “diesel and gas engine”;

- considering (9), please justify the numerical values of 0.25 and 0.4 used for constraining the ESD power, which seems quite small values.

Reviewer 3 Report

The article titled "A Two-stage Optimal Scheduling Model of Micro-grid Based on Chance-constrained Programming in Spot Markets" concerns the subject of uncertainty of wind and solar output  and uncertainty of price from the distribution of this energy.
This manuscript is organized as follows:
- Section 1, this is introduction.
- Section 2,  it establishes micro-grid energy management system integrating wind solar and load storage considering the uncertainty of new energy output.
- Section 3, according to the pilot rules of China’s power spot market operation, the micro-grid participation process is discussed, and a two-stage optimal scheduling model of micro grid based on stochastic chance-constrained programming is constructed with the predicted power of new energy generation and the spot market forecasting electricity price as random variables.
- Section 4, the stochastic simulation technology is used to deal with the random variables in the model, and then a particle swarm optimization algorithm suitable for solving this model is proposed.
- Section 5 introduces the simulation results, verifies the validity of the model and discusses the effect of demand side management and confidence level on scheduling results.
- Section 6 includes the main conclusions of the paper.
The article was prepared very carefully.
The literature is adequate and sufficient.
Reviewer's comments:
- no reference to figure 1 in the text,
- charts are illegible,
- poor quality of charts,
Article raises important issues of the Chinese energy market.
I recommend the paper for publication after suitable corrections I have suggested

Reviewer 4 Report

This paper tackle the micro-grid spot market decision-making issues in term of reliability and economics using two stage scheduling model solved by using PSO. The paper tackles a hot topic and tries to propose a solution for that. However, the authors need to address the following comments;

The author is suggested to explain, what is a novelty in the work as compared to state of the artwork because already some existing work use the penalty factor is real time and day ahead market. Authors are suggested to describe how the proposed solution is better in term of reliability. Authors are suggested to add flow chart to describe the proposed solution In page 6, authors explains the microgrid day ahead scheduling model based on chance constrained programming in order to soften the constraints but did not well explained on which bases we can soften the constraints and how will it effect in real time market if violates. The quality of result images are quite bad, authors should need to improve the quality of all images. In real-time market micro-grid scheduling model, the penalty fee is charged for deviation to MG energy producers, so what is the motivation for them to participate in the market, as renewable energy sources are random and uncertain. Authors are suggested to show what will happen for islanding case, where MG will only be the producer. Authors are suggested to describe how the peak power variation can be control in the proposed framework and what will be the effect on price when renewable energy sources and %SOC of batteries both will be too low to meet the load demand. As the problem is mixed integer programming, so why to use the metaheuristic PSO technique. The PSO algorithm is sensitive to control parameter such as the inertia weight and/or acceleration coefficients. How does authors tuned the values of parameters ? 

Round 2

Reviewer 1 Report

The paper has been consistently improved, both from the literature viewpoint and the clearness of the exposition.

I feel satisfied with the changes, especially for what concerns the simplification of the nomenclature.

Author Response

Thank you for your decent advise.

Reviewer 2 Report

Unfortunately, the authors have not improved the paper as required, as detailed below:

- please add units to each variable reported in the nomenclature section; please check also the correctness of all the definitions there reported, for example Pτ0 is defined “User electricity consumption and price” but it should be just “user electricity price”;

- Figs. 10-11 need to be improved further, namely they are still a little bit fuzzy, numerical values on all the axes should share the same size and font, and the label on one axis is unreadable (objective function value);

- line 176, it seems that “k” should be replaced with “n”, check this issue also throughout the paper;

- Fig. 2 is very useful, but not completely clear, please explain it more extensively within the main text;

- eq. (18) is still unclear, especially the meaning of each term that contributes to the total microgrid income; please explain the meaning of each term by words within the main text immediately after (18), not just the meaning of the symbols there employed; this is also important for better understanding Fig. 2;

- eq. (19), the use of the coefficient 24/ω is still unclear: if ω < 24, this results in amplifying wind and PV powers with no clear reason, please explain which kind of conversion is carried out and why;

- eq. (21) is unclear, especially because the difference between Qon and Qr is not well explained; please address this issue carefully, which is fundamental also for understanding (18);

- Section 4, please define the acronym PSO, eventually in the nomenclature section;

- Section 4 is still unclear, the introduced flowchart is unfortunately not enough informative to clarify how the proposed two-stage procedure works. Please address this issue carefully and extensively, by referring to the proposed algorithm instead of to general stochastic and PSO equations. Additionally, I confirm the need of moving Fig. 12 to this section and of converting it into a subsection of Section 3, as notified to the authors in one of my previous comment that seems to be completely disregarded.

In addition to the previous comments, please find below all my other suggestions that seem to have been completely disregarded (neither replies nor changes have been provided by authors); in addressing them, please consider that line references refer to the original version of the manuscript, so keep it in mind when introducing the required changes:

(major issues)

- Section 5, please report all the numerical values in dedicated tables rather than within the main text (lines 282-304). In this regard, please specify which is the ESD charging and discharging efficiency precisely (about 90% is not acceptable);

- Section 5, please better clarify how day-ahead and real-time market prices shown in Fig. 2 are computed;

- Section 5, lines 324-325, it seems that “power” should be replaced by “electricity” or “energy” as it is given in kWh. In addition, the concept of “electricity/power declared on the day-ahead market” should be explained;

- Section 5, lines 338-339, the planned purchase curve to which these lines refer does not appear neither in Fig. 4 nor in Fig. 5, please clarify. In addition, please quantify and highlight the benefits of real-time market scheduling compared to using just the day-ahead scheduling, which seems marginal, especially in the case of using demand-response program (Fig. 7);

- Section 5, lines 381-391, please summarizes electricity values there reported in a table and explain their meaning.

(minor issues)

- I suggest using Energy Storage System (ESS) and Diesel Generator (DG) instead of Energy Storage Device (ESD) and Diesel Engine (DE) throughout the paper;

- considering (1), the square brackets should be removed (they do not seem necessary) and f should be replaced by “fWT” in order to avoid misunderstanding with (3) and some following equations (for example, (15));

- Section 2.1, please replace “fan” with “turbine” or “WT”;

- considering (3), please define Γ and replace f with “fPV”;

- Section 2.2, line 120, please add “generators” after “diesel and gas engine”;

- considering (9), please justify the numerical values of 0.25 and 0.4 used for constraining the ESD power, which seems quite small values.

In conclusion, please provide a readable file, in which all the changes are clearly highlighted and distinguishable (now two different colours have been used, purple and red, together with many removed text and formula in black); if the track changes mode is used, please provide the doc file along with the pdf. Alternatively, leave just the changes in red, not the removed parts.
